# An Unanticipated Modulation of Cyclin-Dependent Kinase Inhibitors: The Role of Long Non-Coding RNAs

**DOI:** 10.3390/cells11081346

**Published:** 2022-04-14

**Authors:** Debora Bencivenga, Emanuela Stampone, Angela Vastante, Myassar Barahmeh, Fulvio Della Ragione, Adriana Borriello

**Affiliations:** Department of Precision Medicine, University of Campania “L. Vanvitelli”, 80138 Naples, Italy; debora.bencivenga@unicampania.it (D.B.); emanuela.stampone@unicampania.it (E.S.); angela.vastante@unicampania.it (A.V.); myassar.barahmeh@unicampania.it (M.B.); fulvio.dellaragione@unicampania.it (F.D.R.)

**Keywords:** long non-coding RNAs, CDK inhibitors, p16^INK4A^, p15^INK4B^ Cip/Kip CDK inhibitors, p21^Cip1^, p27^Kip1^, p57^Kip2^, ANRIL

## Abstract

It is now definitively established that a large part of the human genome is transcribed. However, only a scarce percentage of the transcriptome (about 1.2%) consists of RNAs that are translated into proteins, while the large majority of transcripts include a variety of RNA families with different dimensions and functions. Within this heterogeneous RNA world, a significant fraction consists of sequences with a length of more than 200 bases that form the so-called long non-coding RNA family. The functions of long non-coding RNAs range from the regulation of gene transcription to the changes in DNA topology and nucleosome modification and structural organization, to paraspeckle formation and cellular organelles maturation. This review is focused on the role of long non-coding RNAs as regulators of cyclin-dependent kinase inhibitors’ (CDKIs) levels and activities. Cyclin-dependent kinases are enzymes necessary for the tuned progression of the cell division cycle. The control of their activity takes place at various levels. Among these, interaction with CDKIs is a vital mechanism. Through CDKI modulation, long non-coding RNAs implement control over cellular physiology and are associated with numerous pathologies. However, although there are robust data in the literature, the role of long non-coding RNAs in the modulation of CDKIs appears to still be underestimated, as well as their importance in cell proliferation control.

## 1. A Short Premise

An unanticipated and surprising breakthrough of the last two decades is the discovery that a large part of the genome is transcribed and what was mistakenly called junk DNA is actually recognized to play crucial roles in determining cell phenotypes and in human diseases. As a matter of fact, the ENCODE (ENCyclopedia Of DNA Element) project has clearly and definitely demonstrated that genes codifying for untranslated transcripts correspond to 60–75% of the human genome [1,2]. This astonishingly wide RNA world is dominated by thousands of lncRNAs (long non-coding RNAs), sequences of more than 200 nucleotides not (generally) transcribed in peptides/proteins. Although a precise quantitative and qualitative evaluation is still lacking, the ENCODE project estimates the occurrence of more than 16,000 distinct types of lncRNAs. On the other hand, different sources report the presence of more than 100,000 lncRNAs [3,4]. Numerous aspects of this large and heterogeneous RNA population have been investigated and several conclusions have been definitely reached. A deep description of lncRNAs is out of the focus of this review since our goal is to describe the regulation of cyclin-dependent kinase (CDK) inhibitors due to this class of molecules and the strict structural and functional interaction observed in some instances to control specific cellular responses. However, some of the lncRNAs features will be described here for the sake of clarity of this review. For a more extensive description, we may address the interested readers to excellent recent Literature revisions [5,6,7,8].

## 2. Long-Non Coding RNAs

Generally, the DNA regions encoding lncRNAs are located in distinct genomic loci including, for the most part, intergenic regions that represent a majority of the human genome. LncRNAs could also result from the expression of sense or antisense transcripts that overlap with specific sequences of protein-coding genes. Usually, RNA polymerase II (Pol II) is responsible for the transcription of lncRNAs, although other RNA polymerases might be employed [9,10]. The sequence of lncRNAs generally includes a classical cap of 7-methyl-guanosine (m7G) at the 5′-end and a polyadenylated tail at the 3′-end. Most intriguingly, mature lncRNAs derive from primary RNAs splicing. Indeed, lncRNA genes might contain (although not in all the instances) exons and introns. The splicing process appears to occur, in several cases, similar to that described for messenger RNAs, including the possibility of alternative splicing [11,12]. As stated above, lncRNAs originate from different genomic regions and might be accordingly categorized in various classes. Different classifications have been proposed and all of them show a clear and intrinsic validity. Here, only for simplicity, we will report an arrangement based on lncRNAs origin and localization.

A number of lncRNAs might result from the transcription, often in an antisense direction, of a region placed near the promoter sequence of a protein-encoding gene [13,14] usually located 0.5–2.5 kbs upstream of the transcription initiation sequence of the contiguous gene. This class of lncRNAs, defined as PROMPT (PROMoter uPstream Transcripts), has generally a short life and nuclear localization [14,15]. Their instability is the consequence of rapid degradation due to the activity of the so-called RNA NEXT (Nuclear EXosome Targeting) complex [16,17]. The NEXT complex is required for the degradation of aberrant and non-coding RNAs, including not only PROMPT but also small nuclear RNAs [18,19].

The function of PROMPT is in large part unclear, although some hypotheses have associated these RNAs with the transcriptional modulation under stress conditions [20]. It is to be underlined that PROMPT genes are localized near their target genes, suggesting a possible direct effect on promoter/enhancer sequences of these genes. On the other hand, they may interact functionally with the close gene target, cooperating to orchestrate an effective cellular response to specific signals [21].

A further class of lncRNAs includes eRNAs (enhancer RNAs) that are significantly longer than PROMPTs, extending for about 2000 bps [22,23]. They are bidirectionally transcribed from enhancers of protein-encoding sequences. In general, eRNAs affect the topology of chromatin structure in regions where promoter/enhancer interactions can occur [24,25,26,27]. This, in turn, suggests that they could specifically modulate transcription. On the other hand, selective removal of these lncRNAs does not frequently affect gene expression, questioning the precise role of eRNAs. Additional studies on eRNAs have demonstrated that the Integrator complex plays a role in the cleavage of the last 30 end-bases of eRNA primary transcripts [28]. The Integrator Complex is formed by at least 14 subunits and has been identified for its interaction with the C-terminus of PolII and for participating in small noncoding nuclear RNAs metabolism [29,30,31]. In experiments where Integrator was depleted, the accumulation of eRNAs bound to PolI was demonstrated. eRNAs, like PROMPTs, are also the target of the exome processing complex [32]. It is of interest to stress that exosome RBP (RNA-binding protein) is largely associated with several RNAs including PROMPTs and eRNAs [32,33]. Numerous similarities between PROMPTs and eRNAs have also been demonstrated, including their short half-life. This is probably related to the exchangeability of promoter and enhancer functions.

A further class of lncRNAs is the so-called lincRNAs (acronym of long intervening noncoding RNAs) [34,35,36]. This class comprises the lncRNAs investigated in more detail. They originate through the transcription of intergenic regions catalyzed by PolII. LincRNAs are similar to messenger RNA in that several of them show an intronic/exonic structure and a m7G cap at the 5′-end and a poly(A) tail at the 3′-terminus [33,34,35,36]. However, lincRNAs contain a number of introns lower than that of mRNAs and have signals of splicing and polyadenylation weaker than those occurring in primary messenger RNAs. Additional features clearly distinguish lincRNAs from mRNAs, including the inability to be translated into protein and being mostly localized in the nucleus [33,34,35,36]. 

In addition to this categorization, a number of alternative classifications have been reported. For example, lncRNAs have been distinguished into intronic lncRNAs, antisense lncRNAs, sense lncRNAs, intergenic lncRNAs, bidirectional lncRNAs, and others. 

## 3. A Survey of Main LncRNA Functions

LncRNAs may regulate a number of cellular processes and particularly gene transcription. It is particularly fascinating that not only the transcript sequence but, also, the lncRNA gene itself (i.e., the DNA sequence) is able to interact directly with other DNA regions thus affecting their topology and, in turn, their activation/repression. The effect might be on neighboring genes or might involve very distant DNA regions. Additionally, lncRNA transcript/DNA interactions could occur and might have effects *in cis* or *in trans*. LncRNAs also modulate the maturation of primary messenger RNAs, their turn-over (i.e., stability/degradation) and their translation efficacy. Finally, lncRNAs are able to interact with proteins and, even, favor or hamper organelle formation. Following, we will briefly discuss some of these mechanisms.

### 3.1. Regulation of Chromatin Structure

LncRNAs regulate chromatin structure interacting with DNA either directly or indirectly, through binding to chromatin proteins. In the first case, lncRNAs form hybrids structure (triple helix, RNA/DNA/DNA, or R loops) [37,38]. The triple helices, whose abundance is not known, might positively/negatively modulate gene expression. For example, the control of transcription of *Sphingosine Kinase 1* (probably due to the interaction of eRNA with *Sphingosine Kinase 1* enhancer) is an example of triple helix formation [39]. In this specific case, the triple helix favors the recruitment of a complex on the enhancer and increases the transcription of the gene. In mESCs (mouse Embryonic Stem Cells), TARID (*TCF21* antisense RNA inducing demethylation), a well-known lncRNA, forms a triple helix with the promoter of *TCF21* (Transcription Factor 21) gene. Then, the triple helix recruits GADD45 (Growth Arrest and DNA Damage 45) protein that, in turn, is recognized by a DNA demethylase (TET1). The demethylation, finally, causes the increased expression of *TCF21* [40,41].

### 3.2. Protein–LncRNA Localization and Function on Chromatin

Numerous pieces of evidence suggest that lncRNAs interact with proteins localized on chromatin. Generally, lncRNAs affect the three-dimensional structure of proteins already able to bind DNA. Several lncRNAs have been definitely identified showing this feature, including ANRIL (Antisense Noncoding RNA in the INK4 Locus), XIST (X-inactive specific transcript), HOTAIR (HOX Transcript Antisense RNA), HOTTIP (HOXA Transcript at the distal TIP), Kcnq1ot1 (KCNQ1 overlapping transcript 1), AIRN (Antisense of IGF2R non-protein coding RNA) and LncPRESS (LncRNA P53-Regulated and ESC Associated 1). While a detailed description of ANRIL and Kcnq1ot1 is reported in different sections of this review, here, we will give a little information on some of the other cited lncRNAs.

XIST is the pivotal trigger of the X chromosome inactivation complex. After its transcription, XIST lncRNA is associated with only one X chromosome. By means of molecular mechanisms whose details are not completely elucidated, XIST acts as a platform for the build-up of complexes that result in the inactivation of a single X chromosome. The XIST capability of inactivating the chromosome in which the XIST gene maps has attracted enormous attention for the treatment of trisomies. Indeed, it could be possible to insert, by gene editing strategies, the XIST gene in one of the chromosomes 21 and inactivating it, reestablishing a normal chromosome 21 balance. This approach has been evaluated as a strategy for Down Syndrome treatment [42,43]. HOTAIR and HOTTIP are able to recruit factors modifying the chromatin structure for modulating *HOXD* and *HOXA* gene clusters, respectively [44,45,46,47]. LncPRESS1 interacts with sirtuin 6, controlling H3 acetylation at Lys56 (H3K56ac) and H3K9ac at the promoter of specific genes [48].

### 3.3. Control of Gene Expression

Several heterogeneous mechanisms explain the ability of lncRNAs to regulate the expression of specific genes. A feature that needs to be underlined is the evolutionarily conserved relationship between a specific lncRNA and its target genes that argues for a non-random interplay/localization and for a specific adaptation. The first effect to be examined is the ability of lncRNAs to silence specific genes or, even, a complete chromosome. As reported before, XIST is a primary actor of one X chromosome silencing [49,50,51,52,53]. Thus, the mechanism of XIST might represent an excellent example of shutting down an entire chromosome. XIST is located in the so-called X inactivation center locus that includes other five lncRNA genes in addition to *XIST* [54,55]. Mechanistically, XIST allows a series of important early or late events. Early events include change (loss/acquisition) in histone modifications due to PRC1 and PRC2 (Polycomb Repressive Complex 1 and 2) and exclusion of RNA PolII [51,56]. Late events are methylation of DNA CpG islands and accumulation on the X chromosome of macroH2A, a histone variant [51,56], which are probably responsible for maintaining the X chromosome inactive state. Other examples of lncRNAs capable of inducing an inactive chromosome condition include ANRASSF1 (Ras Association Domain Family 1 isoform A), COOLAIR, AIRN, CHASERR (CHD2 Adjacent, Suppressive Regulatory RNA), and others [57,58,59,60,61,62].

There are two categories of RNAs that can derive from transcribed enhancers. One class includes the so-called eRNAs that are transcripts of short length, without polyadenylation, unstable, and not maturated by splicing. eRNAs are transcribed bidirectionally [63,64]. The unanticipated discovery that enhancers could be transcribed was made in 2010 but their role still remains in large part enigmatic [22,23]. Moreover, their abundance varies enormously among the tissues [63,64]. Functions associated with eRNAs include enhancer–promoter looping, chromatin-modifying activity, and transcription regulation. The other category includes elncRNA that are generally transcribed starting from the enhancer in a monodirectional fashion. elncRNAs are subject to splicing maturation and undergo to polyadenylation process. The most known elncRNAs include SWINGN (SWI/SNF Interacting GAS6 enhancer Noncoding RNA, also early defined as LINC00565) [64,65], ELEANORs (*ESR1* locus enhancing and activating non-coding RNAs) [66], and BENDR (Bend4-regulating) [67]. SWINGN facilitates the localization of SWI/SNF complex on the *GAS5* promoter sequence as well as of other loci with putative involvement in malignant transformation [65]. BENDR regulates the expression of Bend4 [67]. Importantly, Bend represents a new family of chromatin-binding proteins controlling chromatin topology and modulating transcription [68].

An additional aspect to be underlined is the lncRNA-encoding gene’s ability to modulate gene expression by acting *in cis*. An interesting example is that of the protein-encoding gene *Hand2* (*Heart and Neural crest Derivatives expressed 2*) and two lncRNA-encoding genes, that is, *Upperhand* and *Handsdown*. The *Hand2* protein is a helix-loop-helix transcription factor that plays a pivotal role in heart development including the morphogenesis of ventricular chambers and aortic arch and neural crest-dependent morphogenesis [69,70,71,72,73,74]. *Hand2* genetic changes have been correlated to serious malformations including the tetralogy of Fallot [75]. *Upperhead*, although located near *Hand2*, does not use the *Hand2* promoter but a distinct and bidirectional one. *Upperhand* controls, in cis, the expression of *Hand2* probably by a direct interaction between the DNA sequences [76,77]. Conversely, *Handsdown* promoter forms a loop with Hand2 regulating sequences. The loop is favored by the CTCF (CCCTC-Binding factor) protein. LncRNA *Haunt* shows a different unexpected mechanism [78]. The gene encoding *Haunt* shows an enhancer that activates the expression of *HOXA* genes while *Haunt* transcript inhibits the activity of its own gene enhancer thus inhibiting the expression of *HOXA* genes.

### 3.4. LncRNAs as Scaffold for Establishing Membranless Structures

An outstanding emerging issue, correlated to the connection between lncRNAs and control of gene expression, regards the function of lncRNAs in the formation/structure of paraspeckles. Paraspeckles, frequently defined as nuclear bodies, are cellular organelles lacking a membrane, particularly abundant in the nuclear compartment [79,80,81]. Some lncRNAs appear to represent paraspeckles scaffold components able to recruit peculiar proteins by establishing RNA/protein interactions. Then, these initial assembled structures engage additional proteins by means of protein/protein interactions constructing definite paraspeckles. Such membrane-less organelles are able to link messenger RNAs or transcription-correlated factors thus, finally, modulating gene expression. A central role in paraspeckle formation is played by the lncRNA NEAT1 (nuclear paraspeckle assembly transcript). NEAT1 codifies for two distinct transcripts [82,83]. The two isoforms share a similar 5′-ends but distinct 3′-terminus. The short form presents a poly(A)tail. The long form, NEAT1_2, (about 23 kb in humans) shows a triple helix at the 3′-end [84,85,86,87,88,89]. Apparently, only the long isoform is able to build up paraspeckles. A detailed description of paraspeckle structure and origin is out of the focus of this review but for the great interest in the field, we refer the readers to other recent excellent descriptions [90,91]. Regarding their roles, it is probably related to the sequestration of molecules, either RNA or proteins. This sequestration might reduce the level of specific proteins in the nucleoplasm and decrease their function, for example, at the level of transcription (activation or repression). Alternatively, paraspeckles might favor (or inhibit) miRNA maturation. Several physiological/pathologic conditions have been correlated to paraspeckles levels. These include the response to viral infection, female fertility, positive/negative (depending on the context) effects on carcinogenesis, and nervous system development [90,91].

An important lncRNA associated with nuclear bodies is MALAT1. MALAT1 is the acronym for Metastasis-Associated Lung Adenocarcinoma Transcript 1, which clearly suggests an important role of this lncRNA in carcinogenesis [92,93,94]. Accordingly, MALAT1 is particularly abundant in cancer cell lines arguing for a key role in carcinogenesis. However, distinct *MALAT1* knockout mouse models give contrasting phenotypes that might be explained by the strategies of gene inactivation as well as by the different dimensions of genomic deletions. Human MALAT1 RNA is 8425 nucleotide length that, on the basis of bioinformatic analyses, gives a complex secondary structure able to form 194 helices, 5 probable tetraloops, and 13 pseudoknots. Many intramolecular interactions and internal loops have been also suggested. Functionally, two roles for MALAT1 have been proposed: namely, the regulation of transcription or of alternative spicing (or both). However, mice inactivated for the *MALAT1* gene do not show growth alteration or modified alternative splicing, making a clear understanding of MALAT1 functions still far from being reached.

Since MALAT1 is mostly found in nuclear speckles [95], its depletion causes alterations in the protein composition of these structures, affecting, in peculiar mode, the content of some splicing factors (SON and SRSF2). In brief, MALAT1 might represent a key center for the metabolism of the most abundant RNAs, including U1 small nuclear RNA [96,97,98,99,100]. The function is favored by the localization of MALAT1 on the surface of the speckle (and not in the center as NEAT1). Additional examples in which lncRNAs play the role of the scaffold in the construction of a membraneless structure include small nucleolar RNA-related lncRNAs and 5′ small nucleolar RNA-capped and 3′ polyadenylated lncRNAs [96,97,98,99,100].

### 3.5. LncRNA in Control of Protein Turnover and Cellular Organelles

LncRNAs are able to interact directly with proteins forming lncRNA-protein complexes (lncRNPs). In this way, they might affect, for example, the process of RNA metabolism (splicing and stability) or mRNA translation, controlling protein levels. LncRNAs might also control the activity of transcript splicing by changing the post-translational status of proteins catalyzing messenger maturation. NORAD (Non-coding RNA Activated by DNA damage), a cytosolic lncRNA, is strongly expressed as a consequence of damage to DNA [101]. This lncRNA is able to interact and down-regulate the activity of Pumilio, a protein that increases mRNA degradation [102,103]. Given the remarkable number of Pumilio molecules interacting for each NORAD sequence, the control on the stability of mRNAs exerted by increased levels of NORAD is important. It is also to stress that the activity of Pumilio in cellular metabolism is extremely relevant, acting on the control of cell proliferation and differentiation. Other types of interaction between lncRNAs and proteins might result in a change in protein folding and modulation of key cellular pathways. FAST is an example of lncRNA acting through this mechanism. As a matter of fact, FAST interacts/blocks an E3 ubiquitin-ligase (β-TrCP) responsible for the removal of phospho-β-catenin [104]. An increase in FAST results in the up-regulation of phospho-β-catenin and its subsequent translocation into the nucleus. 

LncRNAs are also able to form complexes with mRNA, limiting their translation, and then, after an interaction with specific proteins, commit the transcript to removal. This has been demonstrated in the case of STAU1 (double-strand RNA-binding protein Staufen homologue 1) [105]. 

Some lncRNAs can interact with miRNA (microRNA) and reduce their level. According to the competing endogenous RNA (ceRNA) hypothesis, lncRNAs may act as endogenous molecular sponges that competitively bind miRNAs when containing specific microRNA response elements (MREs) also present in the mRNA target sequence. In this activity, the quantitative interplay between lncRNA and miRNA is essential, since only an abundance of lncRNA could affect the level of miRNA. An example of this mechanism that has been defined as “miRNA sponge” activity is represented by PNUTS and mir-205 [106]. This micro-RNA is a suppressor of ZEB-1/ZEB-2 and is necessary for correct epithelial cell turnover. The increase of PNUTS reduces the level of mir-205 and up-regulates ZEB1/ZEB2, inducing the EMT phenomena that, as well known, is a pivotal process in metastatization. Alternatively, the lncRNA may compete with miRNA for the same target interaction, thus counteracting the miRNA function. Several examples of lncRNA acting through this mechanism will be described later in this review.

LncRNAs are also capable of localizing in mitochondria and exosomes. In the latter case, following exosome metabolism, lncRNAs are released in the extracellular compartment. Being able to be incorporated into other cells, exosomes might vehiculate lncRNAs into cells where these molecules can exert their action, affecting cell phenotype. An example of lncRNAs that might modulate mitochondrial function is SAMMSON (Survival Associated Mitochondrial Melanoma-specific Oncogenic Non-coding RNA), which controls the maturation of mitochondrial 16S ribosomal RNA [107,108]. Other lncRNAs that affect mitochondria function are lncCyt b, lncND5, and lncND6, which regulate the stability of RNA involved in mitochondria metabolism.

The great heterogeneity of the functions of lncRNA translates into an equally large number of biological effects that range from the control of growth and differentiation to apoptosis or autophagy control and response to different environmental changes and stress stimuli. The physiology of several tissues/organs is modulated by the biological functions of lncRNAs, including hematopoietic and immune, nervous, muscular, cardiovascular, and adipose systems. For brevity, a few examples will be reported. Selected knockout studies in mouse models have implicated a remarkable number of lncRNAs in neuronal differentiation and responses to damage and in particular the lncRNA Silc1 that modulates the transcription factor SOX11. LncRNAs appear also strongly involved in the process of hemopoiesis and in the maturation of the immune system. As a matter of a fact, the expression of genes involved in the immune response appears to be regulated by specific lncRNAs. One key lncRNA is UMLILO identified and characterized in monocytes where it regulates the expression of numerous chemokine genes. Similarly, lnRNAs associated with the survival of numerous hematopoietic cell components (erythroid precursors, macrophages dendritic cells) have been identified. An important interplay between lncRNAs and carcinogenesis has been definitely confirmed in a large number of tumors. Given the role of CDK inhibitors in the development and progression of cancer, the next paragraphs will focus on this key lncRNA role.

## 4. The CDK Inhibitors and LncRNAs

Cyclin-dependent kinases (CDKs) play a vital role in the proper progression through the cell division cycle phases. Timely and phase-specific control of their activity requires the interaction with positive (cyclins) and negative (CDK inhibitors, CDKIs) modulators and peculiar post-translational modifications (generally phosphorylation). Two major classes of CDKIs have been identified, namely the INK4 and the CIP/Kip families. The INK4 family includes four proteins that generally act by competing with cyclin Ds for CDK4/6 binding thus forming an inactive heterodimeric complex made of INK4 protein and CDK4 or CDK6 [109,110,111,112,113,114,115]. On the other hand, the CIP/Kip family comprises three members whose features will be discussed in the related paragraph.

## 5. The INK4 CDK Inhibitors and LncRNAs

INK4 family includes specifically p16^INK4a^, p15^INK4b^, p18^INK4c^, and p19^INK4d^ [109,110,111,112,113,114,115]. Their regulatory roles are incredibly ample and embrace all the major aspects of physiological and pathological phenotypes. These include the modulation of proliferation, differentiation, starvation, cell death, senescence, and molecular bases and the progression of numerous acute and chronic diseases [116,117,118]. Since these proteins are involved in the control of cell growth, particular attention has been paid to their role as tumor suppressors in the process of carcinogenesis and metastasis formation [119,120]. Accordingly, much interest exists in these proteins in cancer therapy developmental studies. We will focus our description on the roles of INK4 proteins, and particularly on INK4A (p16^INK4a^) in tumor development, but we will also take into consideration their relevance in human diseases other than cancer.

The first identified member of the INK4 family, that is, p16^INK4a^, represents a major human tumor suppressor protein. Its encoding gene, *CDKN2A*, is localized on a chromosomal region 9p21.3, frequently deleted in human cancers. The gene, which has a length of about 8.5 kb, is formed by three exons and two introns [110,121]. The protein is composed of 156 residues and shows, in SDS- electrophoresis, an apparent molecular weight of about 16 kDa. p16^INK4a^ inhibits CDK4 and CDK6, thus reducing the phosphorylation of pRb (retinoblastoma protein) and of other members of the retinoblastoma family. This, in turn, prevents entry into the S phase. *CDKN2A* locus, however, encodes for a further completely different protein. In this case, the gene employs a distinct first exon (named exon1β) that has its own promoter and enhancers (Figure 1A). The mature transcript is formed by exon1β/exon2/exon3 presenting a peculiar open reading frame (totally distinct from that of p16^INK4a^) and has a molecular weight, in humans, of about 14 kDa [122,123]. Hence, the protein was named p14^Arf^ (acronym of p14 alternate reading frame). Interestingly, p14^Arf^ interacts with MDM2 (mouse double minute homologue 2) and hampers its activity. Since MDM2 protein represents the E3 ubiquitin-protein ligase involved in the removal of p53, p14^Arf^ accumulation results in p53 increase and in the enhanced expression of p53-controlled genes [124,125]. Therefore, the inactivation of *CDKN2A*/*p14^Arf^* locus has effects on both pRb and p53 and on the two key cellular growth-suppressive pathways.

Loss of function of *CDKN2A* is a common event not only in the initial phases of carcinogenesis but also in the negative evolution of neoplasia. The inactivation of the gene might be due to genetic alterations including either homozygous deletions or point mutation [126,127,128]. In addition, several epigenetic mechanisms controlling *CDKN2A* expression have been described [129,130]. These include various histone modifications, DNA methylation, and regulation by different microRNAs and lncRNAs [131,132,133]. The control of histone methylations involves the activity of proteins participating in PRC1 and PRC2. These complexes mostly maintain chromatin silencing by histone H3 lysine 27 di- and trimethylation [134,135].

Several cancers show genetic/epigenetic inactivation of *CDKN2A*; they include melanoma, pancreatic, head and neck, skin, lung, esophageal, gastric, colorectal, ovarian, prostate, renal cell carcinoma, hepatic cancer, neuroblastoma, hematological cancer, and others [134,136,137,138,139,140,141,142,143,144,145,146,147,148,149,150,151,152,153,154,155,156,157,158,159,160,161,162,163,164,165,166,167,168,169].

Human melanoma and pancreatic carcinoma are tumors in which *CDKN2A* loss of function is extremely frequent [134,136,137]. In particular, *CDKN2A* represents a key melanoma susceptibility gene and its mutations are demonstrated in 20–40% of familial melanoma. The frequency of genetic changes in sporadic melanoma is lower, ranging from 2 to 3%. *CDKN2A* deletion has been evidenced in 50% of cases, while the loss of activity due to mutations has been observed in about 9% of cases. Importantly, the inactivation is associated with an increased possibility of aggressive progression as well as metastasis dissemination. In pancreatic carcinoma, one of the most aggressive (severe) human tumors, *CDKN2A* expression is significantly reduced in about 95% of cases [138,139,140]. The role of hypermethylation in this phenomenon varies between 15 and 25%. Mutations have been detected in more than 10% of hereditary cases. Gastric carcinoma (GC) is a very common cancer type. Hypermethylation of *CDKN2A* promoter has been reported with a high incidence in GC [141,142,143]. This was particularly evident in stomach cancers due to EBV (Epstein-Barr Virus) infection [144]. *Helicobacter pylori* infection is an additional leading cause of this neoplasia. Increased *CDKN2A* methylation is also common in *H. pylori*-dependent GC [145]. Head and neck squamous cell carcinomas (HNSCCs) are additional examples of carcinomas in which *CDKN2A* alterations are frequently detected [146,147,148]. HNSCCs’ localization is heterogeneous in that these tumors might interest pharynx, larynx, and nasal and oral cavities. HNSCCs are typically caused by alcohol and tobacco use and might be due to HPV (human papilloma virus) infection. A high percentage (90%) of non-HPV-dependent HNSCCs show a low expression of *CDKN2A* [149]. This is due to loss of heterozygosity (LOH), mutations, and promoter hypermethylation. Interestingly, also oral leukoplakia shows promoter *CDKN2A* hypermethylation. Alterations of *CDKN2A* are also common in NSCLC (non-small cell lung cancer). The observed frequency is different in distinct studies although it has been demonstrated in all the investigations performed: particularly, in Chen et al., more than 30% frequency was reported [149]. Finally, hypermethylation of the gene has been observed in lymphomas, tumors of the esophagus, colorectal cancers, ovary and prostate neoplasia, and kidney carcinomas [151,152,153,154,155,156,157,158,159,160,161,162,163,164,165,166,167,168,169]. 

Senescence and aging have also been associated with changes in *CDKN2A* expression. In this case, however, an increase in the protein was observed. This phenomenon has been definitely demonstrated and the activation of the *CDKN2A* gene now represents a hallmark of senescence [170,171,172,173]. Accordingly, the genetic locus in which the *CDKN2A* gene is localized constitutes a classical and reproducible hotspot for genome-wide association studies (GWASs) investigating the association of lifestyle-related diseases and age-related chronic pathologies like type 2-diabetes, Alzheimer’s disease, atherosclerosis, coronary artery disease, and other human disorders [172,174,175,176,177]. However, although all these pathologies (including cancers) represent the major causes of death and *CDKN2A* altered expression is indubitably associated with these disorders, the molecular connections between *CDKN2A* function and clinical conditions still need to be clarified. In this context, lncRNA ANRIL seems to play a major role as described in the next paragraph.

### CDKN2A Locus and the Non-Coding RNA ANRIL

The lncRNA ANRIL (Antisense Noncoding RNA in the *INK4* Locus, also named CDKN2B-AS) is one of the major modulators of the *CDKN2A* locus. This lncRNA gene has been identified in a melanoma-affected patient showing a large deletion involving 9p21.3 [178]. Subsequently, data from different laboratories demonstrated that SNPs (single nucleotide polymorphisms) affecting the *ANRIL* gene are not only associated with cancer progression but also play a role in the development of metabolic disorders [179,180,181,182,183,184,185].

As shown in Figure 1A, the *CDKN2A* locus embraces three protein-coding genes that are all transcribed in the same direction [118]. Moreover, the locus contains the *ANRIL* gene that is transcribed in the antisense direction. An additional gene localized not far from the 5′ end of the *CDKN2A* gene codes for the enzyme 5′deoxy-5′-methylthioadenosine phosphorylase (MTAPase) [186]. MTAPase catalyzes the phosphorolytic degradation of 5′deoxy-5′-methylthioadenosine (MTA, a product of polyamine biosynthesis) into adenine and 5-methythioribose-1-phosphate (MTR-1P) [187]. Interestingly, MTA represents the major source of adenine for the recycling of this base, while MTR-1P is an important source of methionine. Structurally, *MTAPase* is localized about 192 kb telomeric to the initiation of *ANRIL* transcription [188]. 

The *ANRIL* gene has a length of about 126.6 kbp and includes 24 exons (Figure 1A). In the first intron of the *ANRIL* gene, a further gene is localized, namely, *CDKN2B*, which codes p15^INK4b^, the second member of the INK4 CDKI family. As reported before, it is transcribed in the antisense direction with respect to *ANRIL*. Numerous isoforms, either linear or circular, originate from ANRIL primary transcript due to alternative splicing [189,190]. Furthermore, the ANRIL RNA sequence embraces LINE, SINE, and Alu repetitive elements. 

The different isoforms present distinct tissue distributions and variable levels [189,190]. Different roles have been ascribed to distinct ANRIL isoforms including: interaction with PRC1 or PRC2, interaction with other RNAs (particularly miRNAs), or functioning as a molecular sponge. From a biological point of view, ANRIL activities include control of growth, programmed cell death, and modulation of cell-to-cell adhesion.

Several mechanisms control the level of ANRIL transcripts, in particular, the activity of its promoter, the specificity of the splicing mechanism, and its metabolism rate (namely its stability). The promoter region of *ANRIL* is shared with that of *p14^Arf^*. In particular, the first exon of *p14^Arf^* and the first exon of *ANRIL* are separated by about 300 bp (Figure 1B) in which a bidirectional promoter is localized [191,192]. This short region allows the transcription of the two genes in opposite directions. The sequence is also rich in enhancer activity. CTCF (CCCTC-binding protein) is a major interactor in this region and its binding occurs when the chromatin is in an active status showing trimethylation of histone H3 on lysine 4 (H3K4) [193,194]. The positive activity of CTCF is unexpected since normally it acts as an insulator [193,194]. CTCF binding is also regulated by methylation of DNA CpG islets. In this case, the methylation hampers CTCF binding and thus, reduces the transcription of both *ANRIL* and *p14^Arf^* genes. Importantly, CpG methylation also decreases/hampers the binding of additional transcription factors on the *ANRIL* promoter, including SMAD3/4, ER (estrogen receptor), and interferon [195,196]. Several further transcription factors modulate ANRIL expression including E2F1, c-MYC, SOX2, Sp1, TET2, and STAT1 (Figure 1B) [195,196]. Since ANRIL might control, among other effects, the expression of genes in the *CDKN2A* locus, its regulation might represent a useful therapeutic strategy for reducing cell growth under a number of pathological conditions characterized by excessive and unrestrained proliferation. In addition to transcription control, three further levels of *ANRIL* regulation should be taken into consideration. First of all, the occurrence of numerous mature transcripts modulates, in a tissue-specific manner, the wide range of ANRIL biological activities. Second, ANRIL levels are controlled by different RNA/RNA interactions, particularly by miRNAs. Third, the different ANRIL isoforms present a specific cellular localization. For example, linear isoforms including exon 1 and, as distal exons, 13b or 19 mainly show a nuclear localization. Conversely, linear or circular isoforms including exons 5, 6, and 7 are mostly localized in the cytosol.

ANRIL RNA assumes a complex secondary structure that strictly depends on its exon composition. The secondary structure includes different local organizations like single- or doubled-stranded RNA, various loop types (hairpin or internal loops), and bulges. Harpin loops show a peculiar relevance in that they present a significant affinity for CBX7 (Chromobox 7, Chromo stands for Chromatin Organization Modifiers) a component of the PRC1 complex.

ANRIL modulates the expression of numerous genes acting in cis or in trans. In cis, ANRIL RNA, during its synthesis, probably positioned by PolII itself at the level of *CDKN2A* locus, interacts with and facilitates (acting as scaffold) the recruitment and organization of the two intricate/variable PRC1 and PRC2. These complexes, as underlined above, catalyze histone 2 ubiquitin ligase activity and histone H3 lysine methylation, respectively. In detail, PRC1 is a family of heterogeneous complexes that, on the basis of its components, are distinguished into canonical and non-canonical complexes. In the case of ANRIL, the recruited non-canonical PRC1 complex is formed by RING1, PHC, CBX7, and BM1, with CBX7 playing the role of lncRNA interactor, while RING1 represents the ubiquitin ligase [197]. Mechanistically, ANRIL interacts with CBX7 and recruits PRC1 on the *CDKN2A* and *p14**^Arf^* genes. The localization of PRC1 is also favored by the interaction of CBX7 with trimethylated H3 lysine 27 (already formed by the activity of PRC2) [198].

PRC2 includes several proteins JARID2, EED, SUZ12, and EZH2 (Enhancer Of Zeste 2 Polycomb Repressive Complex 2 Subunit). EZH2 represents the catalytic subunit able to methylate the lysine 27 of histone H3, forming mono-, di- or trimethylated lysine derivatives. Among these proteins, ANRIL is able to recruit SUZ12 placing the complex in the proximity of the *CDKN2B* gene [199,200,201]. In summary, the function of ANRIL, when acting in cis, is to down-regulate epigenetically the expression of *CDKN2A*, *CDKN2B*, and *p14^Arf^*. This effect is correlated to the chromatin modification due to PRC1 and PRC2 activities. ANRIL is, however, also capable of acting in trans. Again, the mechanistic base of its activity is a chromatin modification due to PRC1 and PRC2 functions. The genes/locus to be regulated are recognized by Alu motifs that allow a precise target identification. Among the target genes modulated through this mechanism, *CARD*8 (Caspase Recruitment Domain Family Member 8) can be cited [201]. CARD8 is a pivotal component of inflammosome and is involved in the activation of the NF-*k*B pathway.

The *ANRIL* gene is overexpressed in numerous cancers, including carcinomas of the prostate, kidney, esophagus, breast, ovary, lung, and stomach, as well as leukemia [202,203,204]. In these neoplasias, generally, the accumulation of ANRIL corresponds to a more aggressive TNM stage. Thus, ANRIL levels are considered an independent factor in the negative evolution of cancer and reduced overall survival [205].

Additional data argue in favor of the view that ANRIL is a pro-oncogenic factor. As a matter of fact, during malignant transformation, DNA damage occurs and, consequently, activation of the ATM/E2F1 pathway is observable. In turn, E2F1 up-regulation results in *ANRIL* increased expression. This last event might negatively affect the carcinogenetic process [206]. Similarly, as noted above, multiple SNPs at 9p.21, associated with human cancers, are specifically localized at the *ANRIL* locus, reinforcing the view that this lncRNA plays a role in human malignancies. It is intriguing that *ANRIL* deletions or translocations have been identified in some neoplasias, including neurofibromatosis and gliomas [207,208]. Furthermore, in melanomas, the occurrence of an abnormal transcript that results from *MTAPase* and *ANRIL* transcripts’ fusion has been found [209].

Overall, these findings point consistently to the possibility that an increase of ANRIL might affect and alter key mechanisms, like a correct cell cycle progression, differentiation, apoptosis, and senescence, resulting in or simply favoring malignant transformation. However, this view is not in agreement with additional observations demonstrating the increase in ANRIL transcript contemporaneously to the accumulation of p16^INK4a^, p15^INK4b^, and p14^Arf^ observed in a large array of cancers [196]. This apparent discrepancy suggests a most complex interaction between the lncRNA and cancers, possibly depending on the neoplasia type.

As anticipated before, the role of the *CDKN2A* locus is not only correlated to carcinogenesis but to several additional human physiological and pathological conditions. In particular consistent GWAS investigations indicate the importance of the genes mapping in this locus in senescence, frailty, type 2 diabetes, obesity, cardiovascular diseases (coronary artery disease, ischemic stroke, abdominal aortic, intracranial aneurysms), atherosclerosis, hypertension, Alzheimer’s disease, multiple sclerosis, glaucoma, endometriosis, periodontitis, and other human pathologies [185,186,210,211,212].

Interestingly, GWAS studies have identified specific SNPs localized at 9p21.3 strictly associated with the risk of developing atherosclerosis and cardiovascular diseases [213,214]. Since these SNPs reside in an apparent desert region (i.e., without protein-encoding genes) but where the *ANRIL* gene is localized (particularly in a region overlapping with 13–19 *ANRIL* exons), much interest has been focused on this lncRNA. The importance of this discovery has been consolidated by the observation that ANRIL is strongly expressed in vascular smooth muscle cells, vascular endothelial cells, phagocytic cells, and atherosclerotic plaque. In vascular endothelial cells, *ANRIL* expression is up-regulated by pro-inflammatory factors and, in turn, increases the levels of inflammasome components (CARD8) or angiogenic factors (VEGF) [215,216]. Additionally, *ANRIL* up-regulation has been reported to enhance the content of serum inflammatory factors. As for smooth muscle cells, ANRIL appears to be involved in the modulation of growth, aging, cell death, and migration [217]. These effects appear mostly related to the ability of ANRIL to control *CDKN2A* and *CDKN2B* expression [212]. Particularly, the lncRNA enhances proliferation by inhibiting *CDKN2B* and/or inhibiting the NF-*k*B pathway [218]. Finally, Alu elements occurring in the ANRIL transcript sequence appear to be able to positively control the capability of mononuclear cells to adhere as well as to proliferate. In summary, a strict correlation between the level of ANRIL, its SNPs on one side, and the phenotype of cells involved in the atherogenesis on the other side exists. A number of clinical investigations have also confirmed this connection. Indeed, several studies have concluded that an altered expression of ANRIL might enhance the negative evolution of coronary heart disease and that modified *ANRIL* transcription and splicing are correlated to the specific polymorphism of a single base [219]. 

Changes in ANRIL isoforms have been also associated with the risk of obesity and the development of type 2 diabetes. GWAS studies have suggested that SNPs interesting *CDKN2A* locus are strictly linked to the development of these important diseases with a large social impact. Obesity and its related metabolic diseases might be considered a global overnutrition disaster due to an altered ratio between the intake and expenditure of energy. Excessive overweight is not only due to an incorrect lifestyle but also to hereditary causes, as suggested by numerous investigations. The relevance of the CDK4(6)/pRb and p53 pathways in adipose tissue development and adipocyte proliferation/differentiation has been clearly defined [220]. Accordingly, loss of function of *CDKN2A* and *p14^Arf^* appear to be involved in obesity and type 2 diabetes. In addition, decreased methylation of the *ANRIL* promoter and the consequent overexpression of such lncRNA have been clearly demonstrated in these energy dysmetabolisms and have been associated with the risk of developing obesity in the adult life [220]. Two important aspects should be underlined regarding the role of ANRIL in obesity and type 2 diabetes. The first is that ANRIL is under the control of ER (as reported before) and ER is a key factor in the regulation of glucose metabolism and tolerance, insulin responsivity, and energy metabolism. The second is that ANRIL regulates the *CDKN2A* locus expression. Thus, the identification of a disequilibrium linkage between *ANRIL* SNPs and altered energy metabolism might be envisioned as a vital discovery and a promising target of therapy in the field of obesity and diabetes treatments.

## 6. CIP/Kip Family Members and LncRNAs

The CIP/Kip (CDK-interacting protein/kinase inhibitory protein) family of CDKI comprises p21^Cip1/Waf1^ (coded by *CDKN1A*), p27^Kip1^ (*CDKN1B*), and p57^Kip2^ (*CDKN1C*). These CDKIs share a common domain for CDK inhibition at their N-terminal end called KID (kinase inhibiting domain) that enables them to bind and inhibit different Cyclin-CDK complexes, such as cyclin E(A)/CDK2, Cyclin A(B)/CDK1 [221,222]. More complex is their functional interaction with the G1 phase-specific Cyclin Ds/CDK4(6) complexes. In this case, in fact, the CIP/Kip proteins, besides being able to inhibit the kinase activity, in certain conditions also play a positive effect in the formation and allosteric regulation of Cyclin D/CDK4(6) complex(es) [223]. These activities strictly control cell division cycle progression and cancer development. On the other hand, the CIP/Kip proteins play important CDK-related and unrelated roles in gene transcription, apoptosis, autophagy, senescence, cytoskeleton remodeling, and cell motility, not necessarily directly related to cell cycle progression, rather affecting cell differentiation, development, and organ biogenesis, and cancer-promoting and progression [222,224].

Studies on cell and animal models have shown that these proteins may have redundant functions, although not all of their activities can be completely superimposed [225,226]. This might be due to specific divergences in the central and the C-terminal domains, and to different mechanisms of level control. In adults, all three CDKIs are found in terminally differentiated cells, with p57^Kip2^ showing low levels and tissue-restricted expression (brain, skeletal muscle, heart, placenta, lungs, kidney, intestine, gonads, with very scarce content in liver and spleen) [227]. Conversely, p21^Cip1/Waf1^ and p27^Kip1^ are more ubiquitously and highly expressed. *CDKN1A* is strongly induced in some cell type-specific differentiation processes (mainly muscle cell differentiation) or is transcribed by p53 in response to genotoxic agents. Hence, p21^Cip1/Waf1^ slows down cell cycle progression and allows cells to exploit the DNA damage repair system. Alternatively, if the repair response fails, cells undergo p53-induced apoptosis [228,229,230]. p27^Kip1^ has a primary role in the cell response to external stimuli, such as cell-to-cell contact inhibition, antimitogenic signals, and nutrient deprivation [231,232]. p57^Kip2^ is the Cip/Kip protein with major involvement in embryogenesis and development, considering that *Cdkn1*c^-/-^ mice present high perinatal mortality and severe developmental alterations and organ malformations [233,234]. Differently, *Cdkn1a*^-/-^ and *Cdkn1b*^-/-^ mice are vital but have increased susceptibility to tumors induced by specific genotypes or by physical and chemical carcinogens and a major risk of age-developed tumors in specific tissues and organs [234,235,236,237,238]. Interestingly, a study from Malumbres’ group has underlined that the controlled expression of p21^Cip1/Waf1^ and p27^Kip1^, and of the INK4A family inhibitors, is also important for proper development in the prenatal life since their loss results in perinatal death and general hypoplasia in different tissues. This appears to be correlated to augmented replicative stress and apoptosis induction in developing tissues, particularly in the nervous system [239]. These observations strongly argue for the need for proper control in the expression of all the inhibitors of CDKs during development and in adulthood, with a particular emphasis on those belonging to the Cip/Kip family.

Complex mechanisms are involved in the spatio-temporal regulation of p21^Cip1/Waf1^, p27^Kip1^, and p57^Kip2^ content, acting at the level of chromatin architecture, gene transcription, mRNA translation, protein post-translational modifications, and degradation. At the level of chromatin modification and mRNA control, lncRNAs play roles in all three CDKIs. Extensively studied and paradigmatic is the lncRNA involvement in maintaining the imprinted state of *CDKN1C* [240]. 

### 6.1. 11p15.5 Locus, CDKN1C, and LncRNAs

The p57^Kip2^-encoding *CDKN1C* gene maps on chromosome 11, at 11p15.5 in humans [241], and on chromosome 7 distal region in mice [242]. Both genomic regions comprise the same cluster of genes subjected to parental imprinting, a process required for proper embryonic development that determines parent-of-origin-specific monoallelic expression. The molecular bases of the genomic parental imprinting process are incredibly fascinating albeit very complex, with differential molecular modifications engendered on parental alleles during gametogenesis and then transmitted to the zygote and to the developing new organism. Recently, it is emerging that lncRNAs play fundamental roles in the mechanism of imprinting establishment [240,243]. This occurs for example for *CDKN1C* and the neighboring genes mapping on human 11.15.5 locus. 

The human 11p15.5 chromosomal locus, which spans about 1 Mb, might be considered the archetype of an imprinted genomic region (Figure 2) [240,244]. It is organized into two domains, one telomeric and one centromeric, that are under the control of two “*in cis*” acting Imprinting Control Regions, ICR1 and ICR2, respectively. ICRs usually show a high density of CpG islets and present a different degree of methylation between alleles, therefore they are also called Differentially Methylated Regions (DMRs) [245,246]. Alterations in both domains and in their imprinting regulation have been associated with important growth and developmental disorders such as Beckwith–Wiedemann Syndrome and Silver Russell Syndrome [246,247,248].

In the telomeric domain, also called the *H19/IGF2* domain, ICR1 acts as a chromatin insulator, separating *H19* and *IGF2* genes and controlling their reciprocal imprinting and expression. *H19* is maternally transcribed in a lncRNA, while *IGF2* encodes for insulin-like growth factor 2, crucial for embryonic growth and development. This gene is paternally expressed [249]. ICR1 is able to regulate reciprocally the interaction of the same pair of enhancers located downstream of *H19* on the promoters of both genes [244,250,251]. Specifically, on the maternal allele, ICR1 is scarcely methylated. Thus, the zinc-finger protein CTCF, that creates topologically domain-associating boundaries, can bind it, hampering the enhancers’ interaction at the *IGF2* promoter. *IGF2* is hence silenced, while *H19* expression is allowed [252,253]. It appears that other proteins, such as cohesin, may participate in maintaining this CTCF-dependent repressive state, probably bringing specific CTCF-recognition sequences together and stabilizing the CTF-mediated DNA loops. Interestingly, cohesin depletion experiments caused disruption of long-range chromatin interactions and changes in the imprinting status of the *IGF2* gene [253].

On the paternal chromosome, methylated ICR1 is not recognized by CTCF and this determines interactions of the enhancers with the IGF2 promoter. Therefore, the reciprocal situation is observed: *IGF2* is expressed and *H19* is repressed. 

The centromeric domain of 11p15.5 locus is approximately 800 kb in size and includes some imprinted genes: *KCNQ1* (Potassium Voltage-Gated Channel Subfamily Q Member 1, also named KvLQT1, KQT-like subfamily member 1); *KCNQ1OT1* (KCNQ1-Overlapping Transcript 1 or KCNQ1-Opposite strand/antisense Transcript 1, also referred as LIT1, Long QT Intronic Transcript 1); *CDKN1C*; *SLC22A18* (Solute carrier family 22 member 18); *PHLDA2* (Pleckstrin Homology-Like domain family A member 2); and *NAP1L4* (Nucleosome Assembly Protein 1-Like 4), all under ICR2 control [227,244]. ICR2, also named KvDMR1, has been mapped inside *KCNQ1* intron 10, encompassing the promoter of *KCNQ1OT* and it is methylated on the maternal chromosome and unmethylated on that of paternal origin [253]. Thus, the paternal *KCNQ1OT1* can be transcribed in the antisense direction with respect to *KCNQ1*, while the maternal allele is repressed [227,254,255].

Importantly, Kcnq1ot1 is a 91 kb long antisense transcript with an exclusive nuclear localization [256,257]. It functions in silencing *in cis* the genes present in the domain, recruiting at the level of their promoter different chromatin modifiers, including *EZH2* and *G9a*, also known as *EHMT2* (Euchromatic Histone Lysine Methyltransferase 2). As reported above, EZH2 is the catalytic subunit of the histone methyltransferase PRC2/EED-EZH2 complex that catalyzes methylation of Lys-9 and Lys-27 of histone H3. These histone changes, introduced at the level of the target genes, lead to a repressive chromatin state. Furthermore, Kcnq1ot1 is able to recruit DNA modifiers as DNMT1 (DNA methyltransferase 1) to its targets’ promoter, determining CpG hypermethylation and reinforcing paternal repression of imprinted genes [258]. 

It has been demonstrated in mice that the KvDMR1 region does not only act in the *in-cis* silencing of the genes mapped in the domain, it also participates in the control of expression of *Cdkn1c* and of the neighbor gene *Kcnq1* from the maternal allele in specific differentiation processes [259,260]. More in detail, Maione’s group, studying myogenic differentiation in a mouse cell model, demonstrated that in non-differentiated cells *KvDMR1* participates, through CTCF, in a repressive long-range chromatin interaction with the *Cdkn1c* promoter [259]. Upon differentiation, the interaction of MyoD, the b-HLH master transcription factor of myogenic differentiation, with a specific KvDMR1 sequence that maps closely to a CTCF binding site, determines the release of this repressive loop and the transcription of the maternal allele. This activity is therefore imprinting-independent [259]. 

The same group also verified that Kcnq1ot1, similarly to KvDMR1, exerts a repressive function not only on the paternal but also on the maternal allele of *Cdkn1c*. Specifically, Kcnq1ot1 is able to bind to an intragenic region of *Cdkn1c* [260]. Importantly, EZH2 and trimethylation of K27 of histone 3 (H3K27me3) accumulated in the same region, in particular on the maternal allele. Upon differentiation, relief of repression is due to MyoD binding to the lncRNA. This was confirmed by the observation that Kcnq1ot1 depletion also reduces EZH2 and H3K27me3 presence in this region, highlighting an additional and unexpected level of p57^Kip2^ regulation in muscle cells [260]. From this description, it appears clear that Kcnq1ot1 is a paradigmatic example of lncRNA that plays a fundamental role in genome imprinting, acting at the level of histone and DNA covalent modification, but it also directs chromatin epigenetic modification on both alleles in an imprinting-independent mechanism for proper lineage maturation and differentiation. The authors also importantly speculated that considering that Kcnq1ot1 is consistently present in almost all tissues where however it shows a variable pattern of chromatin binding and chromatin modulators’ interaction, it is totally conceivable that further, and probably highly complex, levels of interaction/regulation contribute to defining the spectrum of its targets for silencing activity [260]. The other lncRNA codified at the 11p15.5 locus is H19, which is one of the first studied lncRNA subjected to imprinting that have been associated with human disease etiology. Studies in mice have highlighted that it is highly expressed during embryogenesis, but its role in the imprinting of genes as the neighbor *IGF2* has not been clarified and it might be lineage-specific [243] and reference therein. Interestingly, it was shown that expression of *H19* in a transgenic mouse model influences, during embryogenesis, the transcription of a network of at least sixteen imprinted genes (IGN, imprinted gene network) comprising growth modulators as *Igf2*, *Igf2r*, *Cdkn1c*, and *H19* itself. This H19 activity is accomplished in trans, probably by the H19-dependent recruitment of MBD1-H3K9 HMT (Methyl-CpG-Binding Domain Protein 1 H3K9 Histone Methyltransferase) to DMRs and consequent stabilization of repressive epigenetic modifications (trimethylation of H3K9, H3K9m3) on both parental alleles. This activity allows the lncRNA to tune finely the expression of the network genes even distant in the genome [261].

*H19* transcription is strongly reduced at birth, persisting at high levels only in muscle. The role of this lncRNA in myogenic differentiation has been addressed in several studies. Either pro- or anti-myogenic activities have been observed, with multiple mechanisms proposed. In fact, studies reported that H19 acts as a competing endogenous RNA (ceRNA) for the regulation (specifically the downregulation) of the miRNA precursor let-7 RNA, involved in the promotion of myogenic differentiation, or as a scaffold for binding of KSRP (K homology-type splicing regulatory protein), an RNA-processing protein that is involved in lowering the levels of myogenin, the transcriptional factor orchestrating myogenic differentiation. Alternatively, myogenesis can be promoted by miR-675-3p and miR-675-5p, miRNAs that are processed from *H19* exon 1 (Figure 2) [262].

Several pieces of evidence suggest that H19 exerts oncogenic and tumor suppressor activities. It shows high expression in different cancers, including ovarian, breast, bladder, lung, liver, pancreatic, and esophageal [263,264,265], with the level of expression correlated to the poorness of the clinical outcome. The significantly higher risk of congenital and childhood tumors observed in BWS has also been associated with altered *H19*. H19 lncRNA has augmenting effects on proliferation, motility, and invasiveness properties of the cells, while inhibiting apoptosis. However, the precise molecular mechanism is not completely known, and also it has not been understood if this oncogenesis promotion is achieved per se or through the activities of miR-675-3p and miR-675-5p (of which H19 is a precursor), which specifically target tumor suppressor genes such as the retinoblastoma protein Rb, Runx1, and Cadherin 13 [266,267,268]. Further studies are also needed to clarify the tissue- and developmental-phase-specificity of the oncogenic activity of H19.

### 6.2. Additional LncRNAs Affecting CDKN1C Expression in Cancer 

In recent years accumulating evidence has demonstrated that numerous lncRNA genes present an altered expression in cancers. Several of them, including *Tug1* (taurine upregulated gene 1) [269], *Linc00668* [270], *HEIH-coding RNA* [271], *SH3PXD2A-AS1* [272], *SOX21-AS1* (SOX21 antisense RNA 1) [273], might alter *CDKN1C* expression by different mechanisms. Tug1 is a lncRNA mostly expressed in the brain and in the retina, where it may control cell proliferation through epigenetically-driven repression of *CDKN1C* transcription. Tug1 is upregulated in gastric cancer, seemly predicting a negative prognosis [269]. SH3PXD2A-AS1 lncRNA is strongly increased in colon cancer primary specimens and cell lines. The overexpression of the transcript correlated with negative indicators of tumor growth, invasiveness, and metastatization [274]. Mechanistically, SH3PXD2A-AS1 may repress *CDKN1C* expression via EZH2-mediated H3K27me3. Interestingly, rescue experiments demonstrated that overexpression of *CDKN1C* was able to inhibit SH3PXD2A-AS1-dependent carcinogenesis [272]. Linc00511 in non-small cell lung carcinoma [275], SNHG17 in colorectal, gastric cancer, LUCAT1 (lung cancer associated transcript 1), PVT1, also act through a similar mechanism, involving interaction with PRC2/EED-EZH2 complex. Interestingly, LUCAT1 overexpression also represses *CDKN2A* expression [276]. The transcript has been found aberrantly expressed also in clear cell renal cell carcinoma [277], hepatoblastoma, papillary thyroid cancer, and esophageal squamous cell carcinoma. In hepatoblastoma cells, STAT3 plays a role in *LUCAT1* gene transcription, then, in turn, the lncRNA works as a miR-301b sponge by binding competition and causing the upregulation of *STAT3* expression [278]. PVT1-coding gene has been found to be hyper-regulated in several cancers, including ovarian cancers. Interestingly, it has been demonstrated that in OCVAR-3 and SKOV3, cell lines the chromatin at level of the promoter of *PVT1* is highly acetylated (H3K27ac) due to p300 activity [279]. Treatment of the cell line with ketamine, an NMDA (N-methyl-D-aspartate) receptor antagonist widely used as analgesic for cancer pain treatment, decreases the p300-recruitment on *PVT1* promoter, thus inhibiting the lncRNA repressing activity on *CDKN1C* expression [279]. SOX21-AS1 transcript has been found aberrantly expressed in lung adenocarcinoma, associated a poorer cancer prognosis. It may act through the reduction of p57^Kip2^. However, no direct effect on *CDKN1C* was demonstrated. In vitro experiments of *SOX21-AS1* knockdown also abrogate the proliferation and colony formation and causes cell cycle arrest of nephroblastoma cells through p57^Kip2^ upregulation [280]. However, the genes that are the target of SOX21-AS1 repressive activity may be contest-specific, since in human colon cancers the lncRNA affects cancer progression and prognosis through epigenetic silencing of a different Cip/Kip inhibitor, p21^Cip1/Waf1^. On the contrary, in some cases, lncRNAs can activates *CDKN1C* expression. Particularly, LINC00628 is able to bind EZH2 and upregulate p57^Kip2^ level. Interestingly, LINC00628 is down-regulated in human colon cancers and its down-regulation augments the proliferative properties of SW480 and SW620 cells, and inhibits apoptosis. Rescue experiments verified that the overexpression of p57^Kip2^ could reverse the regulatory effects of repressing LINC00628 on the proliferative and apoptotic abilities of CRC. In other instances, the mechanism of action may involve other protein interactions. This is the case of ARHGAP27P1 determines *CDKN1C* expression by interacting with Jumonji-Domain containing 3 (JMJD3), causing demethylation of the *CDKN1C* promoter [281].

### 6.3. CDKN1B and LncRNA

Among the Cip/Kip proteins, p27^Kip1^ is the member mostly regulated at the post-translational level. A plethora of studies have addressed the cellular mechanisms that drive and carry out the timely-controlled degradation of the protein, allowing correct progression of the cell cycle and regulation of the many different biological processes in which p27^Kip1^ is involved. Particularly, p27^Kip1^ undergoes ubiquitin-proteasome degradation in the nucleus in the S/G2 phase. This is due to the activity of the Skp2-SCF ubiquitin ligase complex that specifically recognizes the substrate upon its CDK2/1-dependent phosphorylation on Thr187 [282]. In the cytoplasm, p27^Kip1^, upon translocation from the nucleus at G0-G1 transition, is ubiquitinated by the KPC complex (Kip1 ubiquitination-promoting complex) [283]. Regarding the control of its synthesis, at the transcription level, *CDKN1B* is regulated positively by the transcription factor activities of FOXO and Menin [284,285], while it is repressed by the oncogene Myc. Notably, FOXO is an AKT target. In fact, activated AKT phosphorylates the transcription factor impeding its nuclear translocation and activity [286]. Post-transcriptionally, several microRNAs have been also reported to influence p27^Kip1^ mRNA stability and translation. Moreover, another layer of control is exerted at level of p27^Kip1^ mRNA translation that is directed by an internal ribosomal entry site (IRES) at its 5′-UTR [286]. The IRES is a specialized RNA structure that recruits ribosomes to an mRNA in a cap-independent manner. Several RNA-binding proteins, interacting directly with the IRES, affect IRES-dependent p27^Kip1^ synthesis at the ribosome level [287,288,289]. Among them, heterogenous nuclear Ribonucleoproteins (hnRNPs) and Polypyrimidine Tract-Binding Protein 1 (PTBP1) promote the IRES-dependent translation of p27^Kip1^, while HuR (also known as Elavl1) and its sibling HuD downregulate p27^Kip1^ synthesis [290,291,292]. Interestingly, HuR had been previously identified as a ubiquitous RNP able to bind AU-rich elements in 3′-UTR of its mRNA targets, with a function to stabilize the transcript and increase the protein levels. Among HuR targets, we can annotate cyclin A and B and p21^Cip1/Waf1^, which is specifically induced upon UV damage.

p27^Kip1^ can be considered a haploinsufficient tumor suppressor since monoallelic *CDKN1B* alteration, although rare, has been associated with a major risk of developing tumors and germinal monoallelic p27^Kip1^ mutation causes the multiple endocrine neoplasia, type 4 (MEN4) [293,294]. However, its loss of function in human tumors is mainly due to post-translational alterations driven by the oncogenic activation of fundamental signal transduction pathways such as Src, PI3K, and MAPK [295,296,297]. These deregulated pathways can affect the Skp2-dependent protein degradation and therefore its total cellular levels. p27^Kip1^ cytoplasmic sequestration and loss of nuclear localization, considered an important negative prognostic factor for different human tumors, including colon, prostate, breast cancers. Furthermore, p27^Kip1^ downregulation in human malignancies can be dependent on oncogenic overexpression of miRNAs targeting p27^Kip1^ [298]. The first miRNA acting on p27^Kip1^ to be identified have been miRNA221/222 whose activity downregulates p27^Kip1^ in several human malignancies, such as glioblastoma, prostate, triple negative breast cancers, hepatocellular, and papillary thyroid carcinoma. Other miRNA targeting p27^Kip1^ include miR-196a in cervical cancer [299], miR-24 and miR-24-3p in breast and thyroid carcinomas [300,301,302,303], miR-152-3p in chronic myelogenous leukemia [304]. 

Furthermore, the role of lncRNAs in deregulating p27^Kip1^ levels is rapidly emerging in recent years, although their relevance in cancer development and progression still needs further investigations [305,306,307,308,309,310]. As a matter of fact, their involvement in controlling p27^Kip1^ cellular content in different physiological processes or in pathological conditions other than cancer has also been reported. Among the first reported lncRNAs affecting p27^Kip1^ levels, we can mention Urothelial carcinoma-associated 1 (UCA1), which was originally identified in bladder transitional cell carcinoma but successively found altered in several tumors, such as breast, colon, and glioma. Particularly, in breast cancer cells UCA1 was demonstrated to repress p27^Kip1^ mRNA translation by competitively interacting with hnRNP I [311]. This interaction occurred mainly in the cytoplasm, where UCA1 bound preferentially the phosphorylated isoform of hnRNP I, thus reducing its ability to positively interact with p27^Kip1^ transcript at IRES. The authors also found a negative correlation between UCA1 and p27^Kip1^ in breast cancer tissues by microarray analysis. Successively, this lncRNA has also been demonstrated to repress *CDKN1B* expression in models of HBxAg-induced hepatic tumors, through physical interaction with EZH2 and EZH2-dependent enhanced H3K27me3 methylation at the *CDKN1B* promoter [309]. UCA1 has also been associated with apoptosis evasion and the development of resistance to chemotherapeutic drugs such as platinum and gemcitabine. In bladder cancer, UCA1 was demonstrated to activate the transcription factor CREB that leads to transcription of miR-196-5p. Hence, this miRNA might target the p27^Kip1^ transcript and therefore inhibits cisplatin/gemcitabine-induced apoptosis [309]. Many different targets of UCA1 have been identified through the years. Regarding its mechanism of action, UCA1 has been reported to act as an epigenetic modulator through interaction with EZH2 or with BRG1, a component of the chromatin remodeling complex made by SWI and SNF [312,313,314,315,316,317]. More frequently, it has been shown to act as a sponge for several miRNAs [313,315]. Interestingly, UCA1 gene has 3 exons, and it is transcribed in two main lncRNA isoforms which differ only for the third exon length and for the presence of a 14-nucleotide 5′-extension in the shorter isoform. The roles of the different isoforms have not been clarified, although very recently, the longer isoform, also known as CUDR, has been found prevalently in the nucleus, while the more abundant shorter transcript resulted concentrated in the cytoplasm. This allows to hypothesize a mechanism of epigenetic and chromatin architecture modulator played in the nucleus by the longer isoform, and a miRNA activity regulator exerted in the cytoplasm by the shorter transcript [318].

Yang et al. in 2018 identified a p53-induced lncRNA named TRMP (TP53-regulated modulator of p27^Kip1^) able to negatively influence p27^Kip1^ levels by a mechanism similar to the above-reported UCA1. It competes with p27^Kip1^ mRNA for binding with PTB1, thus repressing IRES-dependent p27^Kip1^ mRNA translation [319]. Hence, TRMP participates in the complex of p53 signaling and in the finely-tuned regulation of the tumor-suppressive activity of p53 in unstressed cells [319], whereas p53 itself stimulates transcription of lncRNA with prooncogenic activities in different cellular context [320]. Interestingly, six splice variants of the codifying gene *RP11-369C8.1* transcript have been identified. Among these, a variant shorter than TRMP and, therefore, named TRMP-S, also exerts pro-oncogenic activities suppressing p27^Kip1^ levels, similarly to TRMP. The mechanism of action is, however, completely different, since TRMP-S binds and stabilizes UHRF1, an E3-ubiquitin ligase, acting at chromatin level through recruitment of DNMT1 and HDAC1 to gene promoters. Importantly, this is one of the first examples of distinct splice variants of the same lncRNA converging on the regulation of the same protein (p27^Kip1^) by means of multiple and distinct mechanisms. 

Other lncRNAs show mechanisms of action involving direct interaction with transcription factors or with proteins acting at level of mRNA translation, thereby modifying mRNA levels or its fate. It has been shown in models of granulosa cells that GCAT1 (granulosa cell-associated transcript 1), also called LINC02690, competes with p27^Kip1^ mRNA for PTBP1 binding. Decreased levels of GCAT1 were associated to G1/S cell cycle arrest [321]. Also, RP1-5O6.5 is overexpressed and considered as negative prognostic factor in breast cancer patients. This lnCRNA promotes breast cancer cells proliferation and invasiveness potential. Mechanistically, it represses p27^Kip1^ mRNA translation through interaction with p-4E-BP1/eIF4E complex, thereby limiting the translational efficiency of p27^Kip1^ mRNA [307]. SOX2OT, that has been recognized as a putative oncogenic RNA in pancreatic ductal adenocarcinoma, can function through direct binding to FUS, a protein controlling both mRNA transcription and translation, thus affecting p27^Kip1^ and cyclin D1 [322]. In prostate cancer models, GAS5 could bind directly to transcription factor E2F1, enhancing the binding of E2F1 to the p27^Kip1^ promoter, and its transcription [306]. 

Several lncRNAs are able to modulate p27^Kip1^ expression epigenetically. Many of them act through physical association with the proteins participating to PRC2 complex (EZH2, SUZ12, EED, and RBAP46/RBAP48). Indeed, it has been reported that approximately 20% of the identified human lncRNAs exert their role by binding to PRC2 and facilitating the recruitment of polycomb-group proteins to the target genes. Among others, it is worth mention: UCA1, as above described; FOXD2-AS1, overexpressed in hepatocellular carcinoma specimens and cell lines [323]; SNHG6, found overexpressed in gastric cancer specimens and cell lines [324]; LINC00511, identified in a lncRNA profile analysis as upregulated in ER-negative breast cancers by transcription factor AP-2 [325]. All these lncRNAs act with a similar mechanism, involving EZH2 binding and p27^Kip1^ repression by promoter trimethylation. The same Kcnq1ot1, amply described for its involvement in *CDKN1C* epigenetic regulation, is transcribed by STAT3 activation and recruits EZH2 to p27^Kip1^ promoter region in a model of spinal cord injury [326]. Finally, MALAT1, in MCL-derived cell lines drives epigenetic silencing of p21^Cip1/Waf1^ and p27^Kip1^ by binding to EZH2, and particularly to a CDK1/2-dependent phosphorylated form of the protein. Consequently, it determines enhanced promoter H3K27 trimethylation. Notably, MALAT1 is found upregulated in several type of cancers, also including MCL mantle cell lymphoma [327], acting through different mechanisms (as discussed in paragraph 3.3). LUADT1 (LUng ADenocarcinoma Transcript 1), identified in lung adenocarcinoma, represses p27^Kip1^ transcription through binding to SUZ12, the core component of the PRC2 complex [328]. The result is, as expected, an increased H3K27 trimethylation at *CDKN1B* promoter. E2F1-induced LINC00668 is overexpressed in human gastric cancers; it has been found in PRC2 complexes and thereby has been considered responsible for the epigenetic repression of several CDKIs, including p15^INK4b^, p16^INK4a^, p21^Cip1/Waf1^, p27^Kip1^ and p57^Kip2^ [270]. 

Paradigmatic is the case of the lncRNA Lockd (lncRNA downstream of Cdkn1b), a polyadenylated lncRNA (AK012387) originally identified as one of several different mouse erythroblast lncRNAs codified by a gene located at 4 kb downstream (3′-) to the *Cdkn1b* gene. Gene ablation of the *Lockd* locus leads to about a 70% reduction of *Cdkn1b* transcription in an erythroid cell line. However, homozygous insertion of a polyadenylated cassette just downstream of the transcription starting site, while reducing by approximately 90% the Lockd transcript presence, had no effect on p27^Kip1^. This elegant experimental strategy demonstrated that *Lockd* controls *Cdkn1b* transcription through an enhancer-like element present in its sequence acting *in cis*, while the lncRNA itself is dispensable. These results also suggest that, particularly when studying the function of ncRNAs, the phenotypes determined by gene ablation (in a cell model but eventually also in animal models) could not be directed correlated to the loss of the lncRNA activity, since they could depend on the deletion of DNA elements (enhancers, for examples) present in the coding sequence. The authors also suggest that, for proving that a specific lncRNA is functional, it would be a more effective and specific strategy to downregulate the z transcript and avoid affecting its codifying genomic sequences [329].

Other lncRNA instead have an effect on p27^Kip1^ levels acting as ceRNA towards miRNAs, with sponge effect. Specifically, the lncRNA human leukocyte antigen complex group 11 (HCG11) positively regulates p27^Kip1^ levels by binding with miR-942-5p and IGF2BP. Interestingly, this lncRNA with oncosuppressive activity is downregulated in osteosarcoma specimens [330]. In a model of liver fibrosis, overexpression of *GAS5* (growth arrest-specific transcript 5) increased p27^Kip1^ protein levels acting as a ceRNA for miR-222, thereby inhibiting the activation and proliferation of primary hepatic stellate cells and fibrogenesis [331]. AC026166.2-001, which appears downregulated in laryngeal squamous cell carcinoma (LSCC) specimens and metastatic neck lymph nodes, is reported to exert a sponging activity on miR-24-3p, thus affecting p27^Kip1^ and cyclin D1 levels [332]. LncRNAs act also in physiological processes or pathological conditions other than cancers. An example might be the transcript of *SNHG1* (Small nucleolar RNA host gene) that was found increased significantly with Parkinson’s disease where it plays a role in autophagy regulation. SNHG1 lncRNA has been reported to interact competitively with the miR-221/222 cluster, thus controlling p27^Kip1^/mTOR signaling and autophagy [333]. Another condition in which p27^Kip1^ is strongly targeted by lncRNAs is the high glucose-induced glomerular hypertrophy observed in diabetic nephropathy. In particular, upon high glucose exposure, mesangial cells show increased p27^Kip1^ levels that cause cell cycle arrest and cell hypertrophy [334]. This cascade of events is generated by STAT3-dependent induction of the lncRNA NEAT1 that, as demonstrated by several experimental approaches, forms double-stranded RNA with miR-222-3p, thereby impeding the miRNA binding to p27^Kip1^ mRNA [334]. Importantly, NEAT1, as reported above in this review, has been identified as a nuclear lncRNA playing a role in scaffolding the paraspekle structures. In addition to the most studied mechanisms of action for the identified p27^Kip1^-affecting lncRNAs, it is important to consider *MIR100HG*, a gene mapping on chromosome 11q24.1, that also hosts in its introns the coding sequences for three miRNAs, including miR-100, miR-125b-1, and let-7a. The transcript corresponding to the lncRNA MIR100HG has been found upregulated in acute megakaryoblastic leukemia [335], in early-stage cervical cancer with poor prognosis [336], and in cetuximab-resistant colorectal cancers [337]. Interestingly, Wang et al. reported that MIR100HG regulated p27^Kip1^ in triple breast cancer cell lines through the formation of an RNA–DNA triplex structure, thus providing a new avenue for lncRNA studies [338]. This interaction on the *CDKN1B* locus may favor the recruitment of epigenetic modulators on the *CDKN1B* promoter to regulate cell proliferation. 

### 6.4. CDKN1A and LncRNA

Human p21^Cip1/Waf1^ is a protein of 164 amino acids codified by *CDKN1A* mapped on the 6p21.2 locus [339]. Like the other Cip/Kip family members, the protein contains the KID in its N-terminal region through which it binds and inhibits several cyclin/CDK complexes, thereby controlling the G1/S phase and G2/M progression. On the other hand, at low doses, like its siblings, p21^Cip1/Waf1^ acts in the assembly of cyclin D-CDK4 complexes and in their nuclear translocation, thus promoting their kinase activity and cell cycle progression [340] and references therein]. Being a direct target of p53, p21^Cip1/Waf1^ is transcriptionally induced in response to DNA damage or oxidative stress, slowing down the cell cycle due to CDK inhibition. A further mechanism to counteract cell division is the inhibition of DNA polymerase delta-dependent DNA synthesis through the PCNA binding domain present in its C-terminal region [341]. Moreover, the protein is also involved in the regulation of gene transcription, apoptosis, senescence, cell stemness, cell motility, and metastatization. Furthermore, direct involvement of p21^Cip1/Waf1^ in DNA repair has also been suggested, although with some controversies [342]. A body of evidence is accumulating showing that p21^Cip1/Waf1^ may work not only as an oncosuppressor, but could also play oncogenic roles, depending on the specific cellular, genetic, and epigenetic context [340]. p21^Cip1/Waf1^ can be regulated by several mechanisms, acting at different levels [340], including: (1) modulated degradation mediated by the activity of multiple ubiquitin ligases; (2) p53-dependent and p53-independent enhanced transcription; (3) transcriptional repressors such as myc; (4) epigenetic regulation, mainly achieved through chromatin H3 and H4 acetylation and DNA hypermethylation at the *CDKN1A* promoter. Also, lncRNAs have been involved in the regulation of p21^Cip1/Waf1^ levels and, as we have seen for the other CDKIs, they are numerous and act by different mechanisms. 

The first lncRNA that is worth mentioning in this context is lincRNA-p21. This is a lincRNA transcribed from a sequence located approximately 15 kb upstream of *CDKN1A*. The 6p21.2 locus where LincRNA-p21 maps is an important site with a high density of lncRNA-codifying sequences. Particularly, Hung et al. [343] recognized five different lncRNAs transcribed from this genomic region. LincRNA-p21 has been first identified as the target of p53 in mouse embryonic fibroblasts and in numerous human cell lines where it mediates p53-dependent transcriptional repression. This activity is exerted by binding to hnRNP-K, a key protein participating in transcriptional repressive complexes, and favoring its recruitment at the promoter of the target genes [344]. Phenotypically, this mechanism of action resulted in the activation of apoptosis and/or in the maintenance of an undifferentiated pluripotent phenotype blocking epigenetic reprogramming (keeping notable levels of H3K9me3 and CpG methylation) at pluripotency genes. Interestingly, gene deletion experiments showed that LincRNA-p21 ablation could reduce the expression of *CDKN1A*, while the lncRNA itself was not acting on the *CDKN1A* promoter. This suggested the possibility that an enhancer-like in-cis element may be present in the *LincRNA-p21* gene. This hypothesis was confirmed by evidencing DNA enhancer elements in the *LincRNA-p21* locus that were directly responsible for the in-cis regulation of *CDKN1A* and other genes [345]. On the other hand, Dimitrova et al. had previously proposed that the lincRNA-p21-hnRNP-K complex works as a coactivator for p53 on the *CDKN1A* promoter to sustain prolonged p21^Cip1/Waf1^ expression [346]. Finally, very recently, a LincRNA-p21 sponge effect on miRNA-17-5p has also been involved in p21^Cip1/Waf1^ increase in conditions of benzene-induced hemotoxicity [347]. 

Among the lncRNAs identified in the *CDKN1A* promoter, Hung and colleagues deeply characterized the p53-induced lncRNA PANDA (p21 associated lncRNA DNA damage activated) located at about 5 kb upstream of the transcription starting site (TSS) of *CDKN1A*. A p53-binding site is present upstream of *CDKN1A* TSS, in the genomic region between *PANDA* and *CDKN1A*. Thus, both genes can be activated by p53 binding and can be transcribed in opposite directions in response to DNA damage. However, the transcriptions of *PANDA* and *CDKN1A* are independent of each other. This is also corroborated by the observation that the p53(R273H) gain-of-function mutation, observed in Li-Fraumeni syndrome, was not able to induce *CDKN1A* and instead maintained the capacity to stimulate *PANDA* transcription. Functionally, PANDA interacts with the transcription factor Y subunit-α (NF-YA), acting as a decoy and detaching it from the target gene promoters. This activity inhibits the expression of pro-apoptotic genes limiting apoptosis and cell senescence in a p53-dependent mode [348]. Finally, the Schmitt team identified a further DNA damage-induced lncRNA whose codifying sequence maps at the *CDKN1A* locus, upstream *CDKN1A* TSS, and also upstream the p53-binding site. It was named Damage Induced Non-Coding (DINO) and was induced by p53, thus being necessary to activate the p53-transcription program. Specifically, DINO interacts with p53, promoting the stabilization of the tetrameric isoform of the protein and thus reinforcing p53 transcriptional activity on many of its target genes, comprising *CDKN1A*. Importantly, DINO is considered a haploinsufficient tumor suppressor. The loss of both (or also of only one) of the alleles determines the inactivation of p53 signaling and tumorigenesis in mice. Furthermore, DINO is recurrently silenced in human cancers through hypermethylation at specific control regions, causing p53 tumor suppression escape [349]. 

Many other lncRNAs affect p21^Cip1/Waf1^ levels. The recently identified LncRNA-ZXF1 transcribed from chromosome 10 and altered in several cancers has been shown to function through two distinct mechanisms: on one side it works as a miR-378a-3p sponge increasing the expression of the miRNA targeting PCDHA3, on the other side it binds to p21^Cip1/Waf1^ preventing the APC/C(Cdc20)-mediated degradation [350].

Other lncRNAs modulating p21^Cip1/Waf1^ levels include HOXA cluster antisense RNA2 (HOXA–AS2), found overexpressed in gastric cancer, affecting the size and stage of the tumor, prognosis, and drug resistance [351]. This lncRNA downregulates *CDKN1A* transcription through EZH2-PRC2-dependent increased H3K27 trimethylation. LncRNAs acting with similar mechanisms include ANRIL, already described, MALAT1, LINC00668, HOTAIR, and others [352]. As noted above, HOTAIR is significantly overexpressed in several human cancers, including cervical cancer, where circulating HOTAIR shows a higher level in patients compared to healthy controls, correlated with tumor aggressiveness [352]. Importantly, downregulation of p21^Cip1/Waf1^ has been mechanistically involved in HOTAIR-dependent cancer radioresistance [353]. Conversely, HOTAIR has also been reported as able to upregulate p21^Cip1/Waf1^ synthesis in ATRA-induced differentiation of AML by sponging miR-17-5p [354].

A further lncRNA affecting epigenetic modification on the *CDKN1A* promoter is FAL1 (Focally Amplified LncRNA on chromosome 1), which was identified in a study that, integrating informatic and functional screening investigations, analyzed lncRNA-coding gene mapping in hot-spots for somatic copy-number alterations (SCNAs) associated with human cancers [355]. Specifically, *FAL1* maps at the 1q21 locus, a focal amplicon that presents copy number and expression altered in ovarian cancer. Mechanistically, FAL1 binds and stabilizes BMI1, the core subunit of PRC1, promoting H2AK119 ubiquitination at the promoter of its target genes, among which one of the most affected is *CDKN1A* [355].

Particularly interesting is the mechanism of regulation of TRERNA1 (Translation REgulatory long non-coding RNA 1), found associated with primary and lymph node metastasis in breast and gastric cancers [356,357]. TRERNA1 was identified as a lncRNA able to inhibit the translation of E-cadherin mRNA [356], and also act at the transcriptional level, influencing the promoter methylation status of its target genes [357]. It has been reported that TRERNA1 is N6-Adenosine methylated; interestingly this m6A modification is the most prevalent RNA change affecting several features of RNAs, including splicing, editing, stability, and activity. Very recently, it has been demonstrated in diffuse large B cell lymphoma that m6A-TRERNA is a substrate of the RNA demethylase ALKBH5 (α-ketoglutarate-dependent dioxygenase alkB homolog 5) and that its demethylated form presents higher stability and therefore higher levels. This ALKB5-dependent increase in TRERNA1 levels has been associated with p21^Cip1/Waf1^ downregulation through EZH2 recruitment on the *CDKN1A* promoter, thus impinging on cell proliferation [358]. 

LncRNAs exerting their regulative role on p21^Cip1/Waf1^ acting as a miRNA sponge include LINC00460, found overexpressed in a variety of human cancers, including head and neck squamous cell carcinoma (HNSCC), where it is able to affect cell proliferation and cell cycle control through a sponge effect on miR-612 and an indirect p21^Cip1/Waf1^ downregulation mediated by reducing the miR-612/AKT2 axial [359]. Also, MIR31HG expression has been investigated and found increased in HNSCCs and oral cancers, where it has been reported as a lncRNA induced by HIF-1α. MIR31HG then interacts with HIF-1α itself, favoring its recruitment on target promoters involved in oral cancer progression [360]. Analyzing tumor specimens, higher expression of MIR31HG correlated positively with HIF-1α itself and negatively with p21^Cip1/Waf1^. p21^Cip1/Waf1^ down-regulation was MIR31HG-dependent, although the mechanism of this modulation has not been investigated [361].

Indirect effects on p21^Cip1/Waf1^ are also exerted by EPIC1, an oncogenic lncRNA whose overexpression is associated with a poor prognosis in luminal breast cancer patients. In in-vitro studies, EPIC1 has been demonstrated to bind and activate MYC, thus enhancing MYC engagement on its target promoters [362]. Through this mechanism, EPIC1 upregulation negatively represses *CDKN1A* expression, thus affecting cell cycle progression. Interestingly, this lncRNA has been identified in a comprehensive characterization of the lncRNA epigenetic landscape in cancer. The study evidenced a general hypomethylation at the level of lncRNA-codifying genes in human cancers, in evident contrast with the epigenetic regulation of protein-codifying genes. Furthermore, the lncRNA epigenetic signature significantly co-occurred with mutations in the TP53 gene in many different cancers. Among the 123 genes that were recurrently epigenetically regulated in multiple cancers, the lncRNA that resulted in being the most frequently epigenetically modified was ENSG00000224271, therefore named (EPigenetically Induced lnCRNA1, [EPIC1]) [362]. Other lncRNAs can affect *CDKN1A* transcription repression via enhancing MYC activity, for example, PVT1 [6]. Finally, the BRAF-activated non-coding RNA (BANCR) positively affects the p21^Cip1/Waf1^ protein, probably by direct binding. This stabilization might be responsible, at least in part, for the oncosuppressive BANCR activity in colon cancer cells [363]. In Figure 3, a model diagram of lncRNA activities on CIP/Kip CDKIs is summarized.

## 7. Conclusions

Undoubtedly, the members of the INK4 and CIP/Kip families are major protagonists in determining a specific cell phenotype. Additionally, the importance of their alterations in human chronic and acute diseases is largely established. Although all the components of these two families show remarkable structural homologies (relatively to the respective family) and a certain level of functional redundancy has been observed, the strict specificity of some of their roles has been clearly demonstrated in that each of them can be only partially replaced by others. Additionally, the ablation of a gene encoding each specific CDKI results in a peculiar phenotype. Numerous mechanisms regulate the level and activity of CDKIs, including transcriptional and translational control, rate of degradation, post-synthetic modifications, subcellular localization, and binding with a variety of interactors. In all instances, the discovery of a novel regulatory mechanism has contributed to clarifying a physiological process or has been associated with important human diseases. An additional layer of modulation has emerged in recent years and is related to the ability of numerous lncRNAs to control the expression of different CDKIs, namely *CDKN2A, CDKN2B*, and *CDKN1C*. Intriguingly, these genes mapped in loci (*CDKN1C*, 11p15; *CDKN2A/2B*, 9p21; *CDKN1A*, 6p21.2) with a high density of lncRNA-codifying genes and where some well-known lncRNA genes are localized, namely *H19*, *KCNQ1OT1, ANRIL*, and *LincRNA-p21*. These lncRNAs are conserved in mammalian species, a finding that indirectly argues for their functional significance. Although a large amount of solid data on lncRNAs has been published, several aspects of these molecules need to be clarified. These include, in particular, their exact relevance to cell phenotype. It is indeed known that a large number of lncRNAs are present in scarce amounts that could not be sufficient for affecting their target activities. The development of animal models ablated for the lncRNA genes (including conditional knock-out) might help in clarifying this vital aspect. On the other hand, the possibility of enhancers underlying the lncRNA-codifying sequence should be taken into consideration. Particularly, a detailed genetic (transcriptome) and protein (proteome) analysis might shed light on the role of specific lncRNAs. On the other hand, as always happens in all fields of biological and clinical sciences, lncRNA research can still be full of unexpected, exciting, and powerful discoveries. 

## Figures and Tables

**Figure 1 cells-11-01346-f001:**
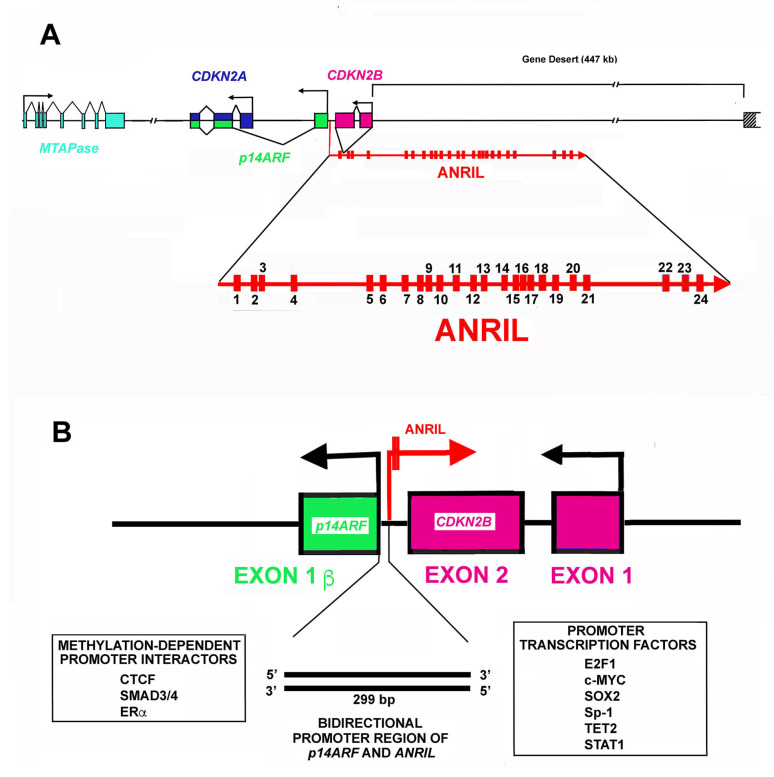
Structure of *CDKN2A/CDKN2B* locus. (**A**). The panel reports the organization in exons (rectangle) and introns (lane) and transcription direction of *CDKN2A*, *CDKN2B*, *p14^Arf^* (p14ARF) and *5′-deoxy-5′-methylthioadenosine phosphorylase* (MTAPase) genes. Note that the *CDKN2A* locus includes two different exons. In particular, exon 1α (blue) is part of the *CDKN2A* gene, while the alternative exon 1β (green) is part of the *p14^Arf^* gene. The panel also shows the *ANRIL* gene structure and its transcriptional direction. As reported, the *CDKN2B* gene is localized in the first *ANRIL* intron. Finally, the figure highlights the gene desert where several enhancers are localized. (**B**). The panel reports the bidirectional promoter of the *p14^Arf^* gene and the *ANRIL* gene. The promoter activity is regulated by several transcription factors listed at the bottom of the panel.

**Figure 2 cells-11-01346-f002:**
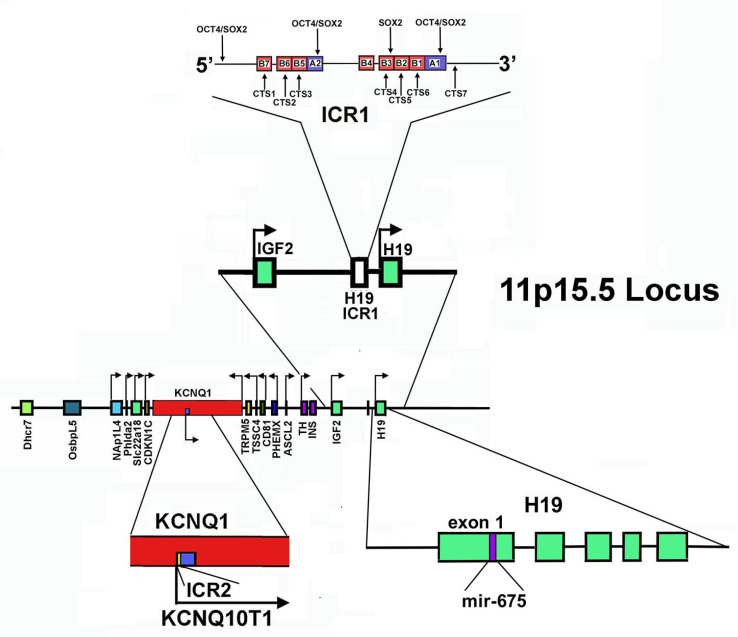
Structure of 11p15.5 locus. The Figure shows the gene mapping at the 11p15.5 locus (in the center). In addition, the direction of transcription is reported. At the top: the localization and the structure of ICR1, as well as the transcription factors interacting with this sequence, are reported. At the bottom: on the left, the organization of ICR2 and *KCNQ1OT1* is shown; on the right, the localization of the mir-675-coding sequence in the first exon of the *H19* gene is evidenced.

**Figure 3 cells-11-01346-f003:**
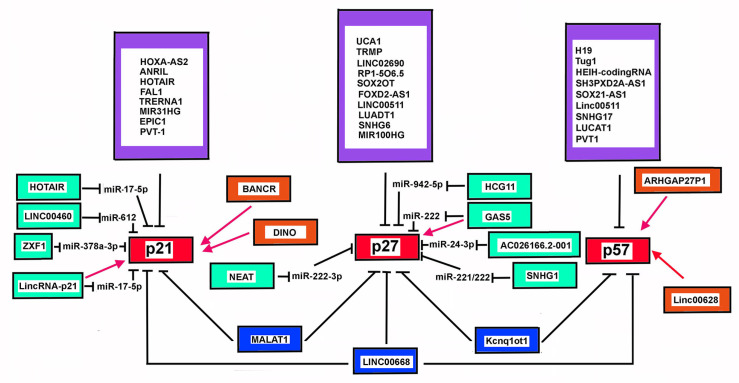
Model diagram of lncRNAs controlling p21^Cip1/Waf1^, p27^Kip1^, and p57^Kip2^ levels. The figure summarizes the lncRNAs affecting p21^Cip1/Waf1^ (p21), p27^Kip1^ (p27), and p57^Kip2^ (p57) cellular contents. Note that the different lncRNAs act by different mechanisms. The purple boxes show the lncRNAs repressing *CDKN1A*, *CDKN1B*, and *CDKN1C* expressions. The red arrows represent lncRNAs that enhance the level of the CKIs. Finally, the green boxes report lncRNAs that prevent the activities of miRNAs, thus up-regulating the levels of p21, p27, and p57.

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
