# Peer review of "An Unanticipated Modulation of Cyclin-Dependent Kinase Inhibitors: The Role of Long Non-Coding RNAs"

_cells, 2022, doi:10.3390/cells11081346_

Round 1

Reviewer 1 Report

This review is aimed to address the modulation of cyclin-dependent kinase inhibitors (CDKIs) by long non-coding RNAs (LncRNAs). The authors introduced LncRNAs first and followed by the general function of LncRNAs. The main content of this review is the summary of two major classes of CKIs and their related LncRNAs. They mainly reviewed the literature of INK4A, one of the four INK4 family members, and all the three CIP/Kip family members, including CDKN1A, CDKN1B, CDKN1C. They have presented a lot of information about these proteins and their LncRNAs. However, it is not clear enough.

  1. As this review is focused on lncRNAs regulating CKIs, it would be better to introduce lncRNA function combined with CKIs related lncRNAs.
  2. Compress the introduction of lncRNA and its functions. It seems there is too much before the main contents.
  3. More small titles are required to guide the readers. Right now, a lot of contents are mixed together. If possible, try to make the review clear and concise.
  4. The main shortage of the manuscript is the lack of a correlation between the lncRNAs and biological processes.

Author Response

We present below a point-by-point response to the Reviewers’ comments (in italic font). We also would like to thank the reviewer for the insightful and constructive comments.

This review is aimed to address the modulation of cyclin-dependent kinase inhibitors (CDKIs) by long non-coding RNAs (LncRNAs). The authors introduced LncRNAs first and followed by the general function of LncRNAs. The main content of this review is the summary of two major classes of CKIs and their related LncRNAs. They mainly reviewed the literature of INK4A, one of the four INK4 family members, and all the three CIP/Kip family members, including CDKN1A, CDKN1B, CDKN1C. They have presented a lot of information about these proteins and their LncRNAs. However, it is not clear enough.

1. As this review is focused on lncRNAs regulating CKIs, it would be better to introduce lncRNA function combined with CKIs related lncRNA

2. Compress the introduction of lncRNA and its functions. It seems there is too much before the main contents.

We thank the reviewer for the suggestions. We compressed the Introduction and highlighted mechanisms of action of lncRNAs relevant for the CKI modulation.

3. More small titles are required to guide the readers. Right now, a lot of contents are mixed together. If possible, try to make the review clear and concise.

According to reviewer's suggestion, we divided some paragraphs inserting some subtitles for sake of clarity.

4. The main shortage of the manuscript is the lack of a correlation between the lncRNAs and biological processes.

We have added brief descriptions of the importance of lncRNA activities in the control of important biological processes.

Reviewer 2 Report

Title: An unanticipated modulation of cyclin-dependent kinase inhibitors: the role of long non-coding RNAs

The authors summarize contributions to the field with an emphasis on ncRNAs. It is a very valuable choice with such a complex system and large number of non-coding RNAs.

Epigenetic changes are an important and highly debated topic whose most often considered tools are DNA methylation, histone modifications and RNA alterations.

The current perspective on ncRNAs has the obvious advantage that epigenetic change can be experimentally induced directly by RNA variations. It covers almost all possibilities of inducing RNA variations.

This is a true-to-field citation review and article description.

However, it might be more efficient if it is more precise and critical in describing the data, because as the data is obtained in a largely heterogeneous system mainly in cell culture, I'm not sure it's always reliable for in invivo studies.

 Just a few suggestions and comments:

Lines 259

MALAT1 (check spelling) is strongly induced in cancer cells but knockout (non-function/or only under certain conditions) non-cancerous in mice! which should be specified.

In case if it is/or not?

Line 661-662

The sentence is not precise, if cohesion participates, what is the function and how?

Line 720

and the resulting epigenetic modifications on the two parental alleles. What type of modification?

Line 928-931

What you mean ? The sentence is unclear because if the DNA fragment only encodes for the ncRNA if it is deleted, one could conclude that the RNA is not transcribed.

Overall, the review is dense but it is a bit difficult for the reader, requires some organization, makes a clear section for imprinted genes and the role of non-coding RNAs in the normal development of pathological cases.

Make it clear what is known in cell culture from in vivo studies.

A table could perhaps facilitate the text, with the references.

We certainly agree that this is definitely the most important direction for future investigators.

Small but important points:

Several punctuations are missing in the text.

Author Response

We present below a point-by-point response to the Reviewers’ comments (in italic font). We also would like to thank the reviewer for the insightful and constructive comments.

The authors summarize contributions to the field with an emphasis on ncRNAs. It is a very valuable choice with such a complex system and large number of non-coding RNAs.

Epigenetic changes are an important and highly debated topic whose most often considered tools are DNA methylation, histone modifications and RNA alterations.

The current perspective on ncRNAs has the obvious advantage that epigenetic change can be experimentally induced directly by RNA variations. It covers almost all possibilities of inducing RNA variations.

This is a true-to-field citation review and article description.

However, it might be more efficient if it is more precise and critical in describing the data, because as the data is obtained in a largely heterogeneous system mainly in cell culture, I'm not sure it's always reliable for in in vivo studies.

According to the reviewer suggestions, we indicated in several instances the model systems used for the experimental studies reported.

Just a few suggestions and comments:

Line 259

MALAT1 (check spelling) is strongly induced in cancer cells but knockout (non-function/or only under certain conditions) non-cancerous in mice! which should be specified. In case if it is/or not?

We sincerely thank the reviewer for raising this point. Accordingly, we have added a discussion about the complexity of MALAT1 gene and transcript, and the proposed mechanisms of action. Furthermore, the phenotypes of knock-out mice for MALAT1 have been reported, underlining the intricacy of the results obtained on genetically modified animals.

Lines 661-662

The sentence is not precise, if cohesion participates, what is the function and how?

We thank the reviewer for stressing the need to clarify this point. We hope that the revised version of the manuscript is clearer than the original submission on this point.

Line 720

and the resulting epigenetic modifications on the two parental alleles. What type of modification?

The information has been added in the revised manuscript.

Lines 928-931

What you mean ? The sentence is unclear because if the DNA fragment only encodes for the ncRNA if it is deleted, one could conclude that the RNA is not transcribed.

We have clarified the point in the revised paper that regards the possible presence of enhancers in loci containing the genes for lncRNAs. In this case, the effect of the gene ablation may not reflect the knock-out of the lncRNA but the enhancer ablation, thus generating confounding results. On this basis,  different strategies of gene editing that impede the transcription of a full lncRNA (for example introducing a polyAA downstream of the transcription starting site) may be preferred. According to the reviewer's request, this point has been explained with more clarity in the revised form of the manuscript,

Overall, the review is dense but it is a bit difficult for the reader, requires some organization, and makes a clear section for imprinted genes and the role of non-coding RNAs in the normal development of pathological cases.

We thank the reviewer for the suggestions. Accordingly, we have divided some paragraphs by inserting some subtitles. We also underlined the role of several ncRNA in human physiology and pathology.

Make it clear what is known in cell culture from in vivo studies.

We have modified the manuscript according to the reviewer indications.

We certainly agree that this is definitely the most important direction for future investigators.

We thank the reviewer for appreciating the topic that we selected for our review article. We strongly believe that it is important to reevaluate some studies on gene deletion in specific genes/locus taking in consideration the concomitant presence of DNA sequences codifying for lncRNA

Small but important points:

Several punctuations are missing in the text.

All the text has been revised, with particular attention to punctuations.

Reviewer 3 Report

The manuscript is an extensive review on lncRNAs regulation of CDKIs. It is largely well written and exhaustive.  I have a few concerns:

1) in section 2 - it would be useful to also comment on whether eRNAs are similarly short lived as compared to PROMPT.

2) in section 3 - there are a lot of examples of lncRNA regulatory mechanisms/functions with no apparent relevance to CDKI regulation.  These examples should be removed as the manuscript title indicates a narrower focus on CDKIs.

3) line 146 - strange use of "tridimensional", 3-dimensional is the common term

4) in section 5, the manuscript describes regulation of each member of CDKI by lncRNAs.  Model diagrams indicating the regulatory pathways of the individual CDKI and lncRNAs would really increase the readability of the review.

Author Response

We present below a point-by-point response to the Reviewer’s comments (in italic font). We also would like to thank the reviewer for the insightful and constructive comments.

The manuscript is an extensive review on lncRNAs regulation of CDKIs. It is largely well written and exhaustive. 

I have a few concerns:

1) in section 2 - it would be useful to also comment on whether eRNAs are similarly short-lived as compared to PROMPT.

According to the reviewer's request, we have added a comment on the short half-life of eRNAs

2) in section 3 - there are a lot of examples of lncRNA regulatory mechanisms/functions with no apparent relevance to CDKI regulation.  These examples should be removed as the manuscript title indicates a narrower focus on CDKIs.

We have simplified this section by reducing the discussion of examples of lncRNAs and their mechanism of action not relevant to the topic of the review.

3) line 146 - strange use of "tridimensional", 3-dimensional is the common term

The word has been modified.

4) in section 5, the manuscript describes the regulation of each member of CDKI by lncRNAs.  Model diagrams indicating the regulatory pathways of the individual CDKI and lncRNAs would really increase the readability of the review.

According to the reviewer's suggestion, a  new figure (Figure 3) has been added to the revised version of the manuscript. It is a model diagram summarizing the activities of the lncRNAs cited in the text on the Cip/Kip proteins p21, p27 and p57.